# Feasibility of 4D-Spatio Temporal Image Correlation (STIC) in the Comprehensive Assessment of the Fetal Heart Using FetalHQ^®^

**DOI:** 10.3390/jcm11051414

**Published:** 2022-03-04

**Authors:** Laura Nogué, Olga Gómez, Nora Izquierdo, Cristina Mula, Narcís Masoller, Josep M. Martínez, Eduard Gratacós, Greggory Devore, Fàtima Crispi, Mar Bennasar

**Affiliations:** 1BCNatal-Barcelona Center for Maternal-Fetal and Neonatal Medicine, Hospital Clínic and Hospital Sant Joan de Déu, 08950 Barcelona, Spain; nogue@clinic.cat (L.N.); nizquierdo@clinic.cat (N.I.); mula@clinic.cat (C.M.); masoller@clinic.cat (N.M.); jmmarti@clinic.cat (J.M.M.); gratacos@clinic.cat (E.G.); fcrispi@clinic.cat (F.C.); bennasar@clinic.cat (M.B.); 2Fetal Diagnostic Centers, Pasadena, CA 91105, USA; grdevore@gmail.com; 3Department of Obstetrics and Gynecology, David Geffen School of Medicine, University of California, Los Angeles, Los Angeles, CA 90095, USA

**Keywords:** fetal echocardiography, speckle tracking echocardiography, STIC, strain, fetal cardiac function, prenatal ultrasound

## Abstract

Fetal Heart Quantification (FetalHQ^®^) is a novel speckle tracking software that permits the study of global and regional ventricular shape and function from a 2D four-chamber-view loop. The 4D-Spatio Temporal Image Correlation (STIC) modality enables the offline analysis of optimized and perfectly aligned cardiac planes. We aimed to evaluate the feasibility and reproducibility of 4D-STIC speckle tracking echocardiography (STE) using FetalHQ^®^ and to compare it to 2D STE. We conducted a prospective study including 31 low-risk singleton pregnancies between 20 and 40 weeks of gestation. Four-chamber view volumes and 2D clips were acquired with an apex pointing at 45° and with a frame rate higher than 60 Hz. Morphometric and functional echocardiography was performed by FetalHQ^®^. Intra- and interobserver reproducibility were evaluated by the intraclass correlation coefficient (ICC). Our results showed excellent reproducibility (ICC > 0.900) for morphometric evaluation (biventricular area, longitudinal and transverse diameters). Reproducibility was also good (ICC > 0.800) for functional evaluation (biventricular strain, Fractional Area Change, left ventricle volumes, ejection fraction and cardiac output). On the contrary, the study of the sphericity index and shortening fraction of the different ventricular segments showed lower reproducibility (ICC < 0.800). To conclude, 4D-STIC is feasible, reproducible and comparable to 2D echocardiography for the assessment of cardiac morphometry and function.

## 1. Introduction

Fetal echocardiography has dramatically improved in recent decades. The 2D modality has been mainly used for congenital heart disease (CHD) diagnosis [1,2,3]. However, advances in fetal imaging and technology, with the incorporation of new modalities such as M-mode [4], tissue Doppler [5], 4D-Spatio-Temporal Image Correlation (4D-STIC) [6,7] and, more recently, speckle tracking echocardiography (STE) [8,9,10] have led to an improvement in fetal cardiac evaluation, not only in CHD detection, but also to carry out a comprehensive evaluation of cardiac morphometry and function [11]. By tracking the endocardial border, STE allows one to evaluate myocardial deformation and global and segmental biventricular morphometry and function. Although STE has been widely used in pediatric and adult cardiology [12,13,14], its application in fetal life is scarce [15,16,17,18,19,20,21,22], probably due to the difficulty in adapting adult-designed software to prenatal cardiac evaluation [23]. On the one hand, fetal echocardiography has its own constraints including limited access and small size of the fetal heart, fetal movements, high heart rate and technical aspects in image acquisition. On the other hand, STE has specific and inherent limitations such as variability of reference values when using different equipment, nonsystematic image settings and processing (spatial resolution, frame rate), and the use of different software’s [24,25].

In this context, fetal specific STE software Fetal Heart Quantification (FetalHQ^®^) was recently developed as a promising offline tool that permits the study of global and regional (24-segment) ventricular shape and function [26,27,28,29,30,31,32,33]. Some groups have reported the application of this novel technology to the study of CHD [34,35], and fetal cardiac adaptation to different fetal conditions (fetal growth restriction [36], diabetes mellitus [37], twin-to-twin transfusion syndrome [38]). Despite its potential for further studying the fetal heart, its applicability is limited given the requirement of a high resolution four-chamber view in a specific angle of insonation. The use of 4D-fetal echocardiography (4D-STIC) permits the acquisition of cardiac volumes for offline analysis, also with high frame rate and good resolution and the possibility of postprocessing modification of parameters such as the angle of insonation, the cardiac plane, and the frame for evaluation [7,23,39]. However, no previous studies have specifically compared 2D- and 4D-STIC STE using FetalHQ^®^ for morphometric and functional assessment of the fetal heart. 

The aim of this study was to evaluate the feasibility and reproducibility of 4D-STIC STE by FetalHQ^®^ for morphometric and functional parameters in healthy fetuses and to compare it to 2D STE.

## 2. Materials and Methods

### 2.1. Study Design and Participants

Prospective study including 31 singleton pregnancies from 20 + 0 to 40 + 0 weeks of gestation attended at the Maternal-Fetal Medicine Department of BCNatal (Hospital Clínic and Hospital Sant Joan de Déu) between September 2020 and January 2021. Low-risk pregnant women were eligible and invited to participate in the study. Exclusion criteria were age <18 years old, ultrasound or chromosomal anomalies and maternal and fetal conditions with known cardiovascular impairment such as diabetes or hypertension, antiretroviral treatment and fetal growth restriction (fetal growth <10th centile according to local standards [40]). Baseline and perinatal data were obtained from the medical records. Gestational age (GA) was calculated according to crown-rump length in first-trimester ultrasound. All participants underwent a single fetal standard ultrasound and echocardiography, performed by two expert sonographers (L.N. and O.G.), to exclude cardiac or extracardiac anomalies following recommended guidelines [41,42]. Estimated fetal weight was calculated according to Hadlock et al. [43], in cases in which it was not available in the two weeks prior to the fetal echocardiography. Fetal weight centile was calculated according to local references adjusted by GA and fetal gender. Doppler pulsatility indices of umbilical, middle cerebral artery and ductus venosus, as well as maximum systolic peak velocity of the middle cerebral artery, were also evaluated.

Study protocol was approved by the Ethical Committee of the institution (Reg. HCB/2019/0540), and written consent was obtained from all participants.

### 2.2. Fetal Echocardiography

Fetal cardiac 2D clips and 4D-STIC volumes were acquired using a Voluson E10 (GE Healthcare Ultrasound, Milwaukee, WI, USA) with a C2-9D convex probe (3–9 MHz) and RM6C convex matrix-array volume probe (2–6 MHz), respectively. All cardiac images were acquired using Speckle Reduction Imaging (SRI) 3 and Compound Resolution Imaging (CRI) 2 and stored in 4D View (GE Medical Systems, Milwaukee, WI, USA) for offline STE analysis.

Two-dimensional 4-chamber clips were obtained at an apical 4-chamber plane with an angle of insonation between the ultrasound beam and the interventricular septum of 45° ± 20°, including at least three complete heart cycles without maternal and fetal movements. Acquisition was made with a frame rate higher than 60 Hz and adequate zoom so that the thorax filled most of the ultrasound screen [20]. 

Cardiac volumes were acquired preferably in the same projection as 2D clips. The acquisition of 4D-STIC volumes was standardized as follows: acquisition time of 7.5 s, angle range of 20–30° and no fetal or maternal movements. Prior to STE evaluation, 4D-STIC volume was adjusted to obtain a perfectly aligned 4-chamber view according to the following steps: 1. The reference dot was placed at the crux cordis in the A plane of a multiplanar view. 2. The Z-axis was rotated until the apex was placed at 0°. 3. The X and Y axes were rotated to systematically obtain an improved 4-chamber view (Figure 1). 

### 2.3. Speckle Tracking Analysis

Two dimensional 4-chamber clips and 4D-STIC cardiac volumes were loaded onto FetalHQ (BT20, GE, Medical Systems). M-mode trace obtained across lateral right ventricle wall at the level of tricuspid annulus was used to define a single cardiac cycle by identifying end-systole and end-diastole, as previously described [20]. The septal, the lateral atrioventricular (AV) valve annulus and the apex of each ventricle were manually identified in the previously defined end-systole frame. Endocardial border was tracked semi-automatically, obtaining a speckle-tracking algorithm, along the cardiac cycle (Figure 2). End-diastolic endocardial tracking was then adjusted if necessary, especially the RV, so that the endocardium, the muscular trabeculations and the moderator band were considered the RV cavity [44]. Both ventricles were divided into 24 segments automatically [27] to allow analysis of the base (segments 1–8), mid ventricle (segments 9–16) and apex (segments 17–24). 

Biventricular global longitudinal strain (GLS) and area derivate as a function of time derivate graphs after the analysis are displayed (Figure 3). The end-systolic frame was adjusted in the area derivate as a function of the time derivate graph at the time the function crossed 0 on the Y axis. 

FetalHQ^®^ analysis automatically calculated the global and segmental cardiac morphometric parameters, which included biventricular end-diastolic areas and lengths and LV volumes, as well as the transverse and longitudinal diameters and sphericity indices (SI), for 24 defined segments of both ventricles as described by DeVore et al. [27,32]. Global cardiac function was also assessed by calculating biventricular fractional area change (FAC) [30] and LV ejection fraction (EF), stroke volume (SV) and cardiac output (CO) [28,45]. Since the measurement of estimated fetal weight in the same scan is necessary to obtain the result for CO, CO was only assessed in 24 fetuses, unlike the other the parameters, which were assessed in 31 fetuses. Additionally, biventricular GLS [31,46] and the fractional shortening of the 24 previously defined ventricular segments were also calculated to evaluate biventricular longitudinal and radial function [26], respectively. The results of the analysis were exported as a comma-separated values (CVS) file and converted to an Excel spreadsheet (Microsoft Corp., Redmond, WA, USA).

All measurements were performed offline by three expert Fetal Medicine operators (LN, MB and OG) for interobserver reproducibility and to compare 2D- and 4D-STIC modalities. A second analysis by a single observer (LN) was performed at least 1 month after the first measurement for the intraobserver reproducibility. All operators had a learning curve of more than 20 fetuses prior to the study.

Additionally, new FetalHQ^®^ software (BT21, GE, Medical Systems) with the introduction of Quiver [47], a new tool specially designed to enhance the identification of the septal and lateral atrioventricular annulus was used to reanalyze 10 cases and to compare with no Quiver tool.

### 2.4. Sample Size Calculation

The sample size required for reproducibility analysis (30 fetuses) was calculated following Bonnet’s et al. formula [48].

### 2.5. Statistical Analysis

Statistical analysis was performed using IBM SPSS Statistics for Windows statistical package (version 25, IBM Corp., Armonk, NY, USA). 

Inter- and intraobserver reproducibility of STE using 4D-STIC was assessed using Intraclass Correlation Coefficients (ICC) and their 95% confidence intervals using a two-way random model. In addition, ICC was used to assess reproducibility of STE using 2D vs. 4D-STIC. Inter- and intraobserver reproducibility of the 4D-STIC STE using Quiver tool was calculated using ICC. Agreement between operators was studied with the Student t test. Limits of agreement, standard error and 95% limit of agreement were calculated and Bland–Altman plots were obtained. 

## 3. Results

### 3.1. Characteristics of the Study Population

Maternal and perinatal characteristics of the study population are described in Table 1. Median GA at ultrasound was 28.3 weeks (20.3–39.3 weeks) (20 patients between 20–30 weeks and 11 between 30–40). Fetal ultrasound showed a mean estimated fetal weight of 1428 ± 663 g with a mean centile of 51 ± 31.3. Normal umbilical and fetal Doppler parameters were confirmed in all cases.

### 3.2. Feasibility

Adequate speckle tracking analysis was achieved in all 4D-STIC volumes and in all except one 2D clip due to the poor delimitation of the right ventricular cavity. A frame rate above 60 Hz was achieved in all cases with a mean frame rate of 80 Hz in 2D clips and 107 Hz in 4D-STIC.

### 3.3. Reproducibility

The 4D-STIC intraobserver and interobserver reproducibility ICC for the most relevant FetalHQ^®^ parameters is detailed in Table 2, Appendix A and Table 3, Figure 4, respectively. Our results show excellent reproducibility (ICC > 0.900) for global morphometric parameters, including biventricular areas, longitudinal, midventricular and apical diameters. Reproducibility was also good (ICC > 0.800) for biventricular basal diameters. On the contrary, the study of the SI of the different ventricular segments showed poor reproducibility. The repeatability of global functional parameters was also good. LV GLS and LV volumes showed excellent intraobserver reproducibility, while RV GLS, biventricular FAC and LV EF and cardiac output demonstrated good intra, and interobserver reliability. Nevertheless, biventricular FS showed a lower reproducibility, especially FS of the basal segments of both ventricles. No statistically significant differences between operators were observed neither systematic bias for the studied parameters.

Comparison between 2D- and 4D-STIC echocardiographic modalities showed similar data (Table 4, Figure 5). The repeatability was good for global biventricular morphometric and functional parameters and poor for the SI and SF of the different ventricular segments, especially for the segments corresponding to the base of both ventricles. The best concordance was found for biventricular areas and LV volume. Again, no statistically significant differences between operators were observed neither systematic bias for the studied parameters.

### 3.4. Subanalysis Using Quiver Tool

The subanalysis performed in 10 cases using the Quiver tool demonstrated a slightly better reproducibility for most global morphometric and functional parameters but again low reproducibility for the SI and SF of the different biventricular segments, both using 2D- and 4D-STIC. Additionally, ICC and 95% confidence intervals of the Quiver tool analysis as well as the comprehensive study of the 24 biventricular segments are provided in the Appendix A.

## 4. Discussion

This study first demonstrates that 4D-STIC STE is feasible, reproducible and comparable to 2D STE when assessing global cardiac morphometry and function using FetalHQ^®^. Our results show excellent reproducibility for most of the global cardiac morphometry and function variables evaluated, but worse results for segmental analysis of SI and SF. 

### 4.1. Speckle Tracking Echocardiography Using 2D- and 4D-STIC

Since the first use of 2D STE in fetal cardiology in 2008, only a few publications have focused on the analysis of biventricular GLS on healthy fetuses, reporting a good reproducibility [49,50]. However, there are still limited data on GLS normality ranges throughout the pregnancy [16,22,31,51,52,53,54] and its application in different clinical scenarios. In fact, more recent studies, which have incorporated the latest technological advances, point to the need to better evaluate STE reproducibility and its correlation with other morphometric and functional echocardiographic parameters before introducing STE into clinical practice. In recent years, FetalHQ^®^ has shown excellent reproducibility for the analysis of morphometric and functional cardiac parameters [26,27,30,32,55,56,57,58]. FetalHQ^®^ automatically defines 24 segments of the ventricles and provides information of the shape (SI) and radial systolic function (SF) by the calculation of the longitudinal and transverse diameter of each segment. Although the reproducibility of this segmental analysis was initially reported to be very good [26,27], more recent studies have reported worse data [55,57]. Additionally, none of the previous studies have reported results on the reproducibility conducting a comprehensive study of global and regional cardiac morphometric and functional data. 

Focusing on 4D-STIC, the only group that had applied 4D-STIC for STE study in fetal life was Dodaro et al. The authors analyzed the feasibility and reproducibility of LV function evaluation in a cohort of fetuses between 20 and 40 weeks of gestation, reporting moderate interobserver (0.562) and good intraobserver (0.857) agreement for LV GLS and moderate interobserver (0.544) and intraobserver (0.647) repeatability for LV EF [59]. Our data showed better results on LV GLS and EF reproducibility using 4D-STIC STE (Table 2 and Table 3) which could be explained for different reasons. First, we defined a standardized protocol both for the acquisition of cardiac volumes and for the subsequent processing in order to always carry out the offline cardiac evaluation in the same way. Second, this strict protocol allowed us to obtain all cardiac volumes with a very high FR (mean of 107 Hz), which was even higher than that achieved with 2D STE (80 Hz). Third, our study was conducted by experienced fetal medicine specialists with previous expertise in 4D-STIC, while e-STIC evaluations were performed by less experienced sonographers with only three weeks training on FetalHQ^®^. Finally, we evaluated a greater cardiac morphometric and functional parameters comparing 4D-STIC and 2D STE, being able to demonstrate that both techniques are reproducible, with the best performance for global cardiac morphometric and functional parameters and the worst for segmental cardiac shape analysis (SI) and radial function (FS). 

### 4.2. Fetal Cardiac Morphometric and Functional Assessment Using 4D-STIC STE

To our knowledge, this is the first study to assess 4D-STIC STE feasibility for a comprehensive morphometric and functional cardiac evaluation and to compare its reproducibility to 2D STE. Our 4D-STIC results showing a good reproducibility for global cardiac morphometric evaluation are in accordance to most previously 2D STE published data, which supports the use of FetalHQ^®^ with both modalities. On the contrary, the available studies on 2D STE reproducibility for segmental cardiac morphometric evaluation show discrepant data. While some recent studies [55] have reported a good reproducibility for biventricular SI (ICC > 0.758), other studies such as ours have shown poorer results, especially for assessing the shape of the basal segments of both ventricles [57]. We hypothesize that manual adjustment applied for improving the delineation of both ventricles, especially the RV, led to differences in the endocardial delineation, in particular variations of transverse diameters that compute for the SI and FS segmental analysis. In our study, we performed a manual adjustment of the semi-automatic tracking of both ventricles in most of our cases, especially for the RV, considering the endocardium and moderator band as part of the ventricular cavity [44]. Furthermore, SI and FS are calculated by a mathematical formula (SI: end-diastolic longitudinal diameter/end-diastolic transverse diameter, for each segment; FS: (end-diastolic transverse diameter-end-systolic transverse diameter)/end-diastolic transverse diameter, for each of the 24 segments), which may increase the error of both measures and explain the poorer reproducibility compared to other parameters that do not apply formulas. Finally, different orientation of the four-chamber view acquisition (apical vs. transverse) between studies could also explain discrepant results. It is important to note that due to lateral resolution echographic properties, some of the echoes of the lateral walls and the upper part of the septum can be missed when analyzing an apical compared to a transverse four-chamber view. Further studies are necessary to better define these methodological issues and their impact on STE reproducibility of biventricular segmental morphometric evaluation [23].

With respect to functional heart assessment, our study is the first one that has so far included the most extensive analysis of the reproducibility of a large number of cardiac functional parameters using both 4D-STIC and 2D STE. We demonstrate a good 4D-STIC STE reliability for biventricular systolic function assessment, with better performance for LV systolic evaluation, including both GLS and FAC, in comparison with the RV. This can be explained by the more complex 3D structure of the RV. Regarding LV systolic parameters (LV volumes, EF and CO), we also report good reproducibility. CO ICC was the lowest regarding LV systolic function, but we only included 24 cases to assess the reproducibility of this parameter, since evaluation of the estimated fetal weight in the same exploration is necessary to calculate CO. On the other hand, biventricular SF showed a poor reproducibility, a finding that has also been reported by other groups [55,56] and that correlates with the poorer reproducibility that we also found for the SI of the different segments of both ventricles. Again, differences in four-chamber-view orientation, methodology and manual tracking could explain our poorer reproducibility results. 

### 4.3. Subanalysis Using Quiver Tool

To better understand our poor results on the segmental ventricular evaluation using fetal HQ, we evaluated 10 cases using Quiver technology, which is supposed to facilitate the identification of the septal and lateral AV valve annulus by displaying two frames before and after end-systole and end-diastole frame [47]. However, our study failed to demonstrate a better performance with Quiver. We are aware that the analysis was carried out in a relatively small number of fetuses, and this technique may need a learning curve to obtain better results; furthermore, more studies are needed to validate previous studies on this technology.

### 4.4. Strengths and Limitations

We report a rigorously defined acquisition protocol both in 2D- and 4D-STIC modalities defining clear anatomic landmarks postprocessing analysis and recommendable frame rate, which allowed us to obtain a good reproducibility for most of the echocardiographic parameters studied. We also included fetuses ranging from 20 to 40 weeks of gestation to validate the results in both second and third trimesters. Moreover, the comprehensive evaluations of the fetal heart we perform in our study allow us to compare the reproducibility between global and segmental cardiac parameters and to identify our worse results for the segmental analysis. On the other hand, this study also has some limitations. Firstly, we are aware that the sample size of our study could be a weakness, especially for the assessment of CO, as we only included 24 fetuses. Secondly, although 2D STE has been previously validated with normal and abnormal hearts, we have only included in the study healthy fetuses. Thus, more studies would be necessary in order to confirm our results in fetuses with cardiac or extracardiac anomalies. Finally, we are aware that all of our 2D clips and 4D-STIC volumes are in an apical oblique four-chamber view that can be difficult to achieve in the third trimester, or in other fetal conditions such as oligohidramnios. DeVore et al. described that apex-down four-chamber views are eligible indistinctly from apical views for 2D-STE [20], but more studies using basal four-chamber view 4D-STIC volumes should be carried out in order to validate their results.

## 5. Conclusions

To conclude, we confirm that 4D-STIC STE is feasible, reproducible and comparable to 2D echocardiography for the assessment of global cardiac morphometry and systolic function, including GLS. Although it requires a learning curve, the results of this study are encouraging in using FetalHQ^®^ in future studies to assess fetal cardiac remodeling in different maternal and fetal conditions. If our results are confirmed, 4D-STIC would allow not only structural evaluation of fetal cardiac anatomy but a comprehensive structural and functional evaluation by a unique cardiac volume acquired in the four-chamber view. This would enable the use of FetalHQ^®^ for telemedicine. Future technical improvements in the semi-automatic tracking of the endocardium—to avoid manual correction—are warranted to improve reproducibility of segmental fetal cardiac evaluation.

## Figures and Tables

**Figure 1 jcm-11-01414-f001:**
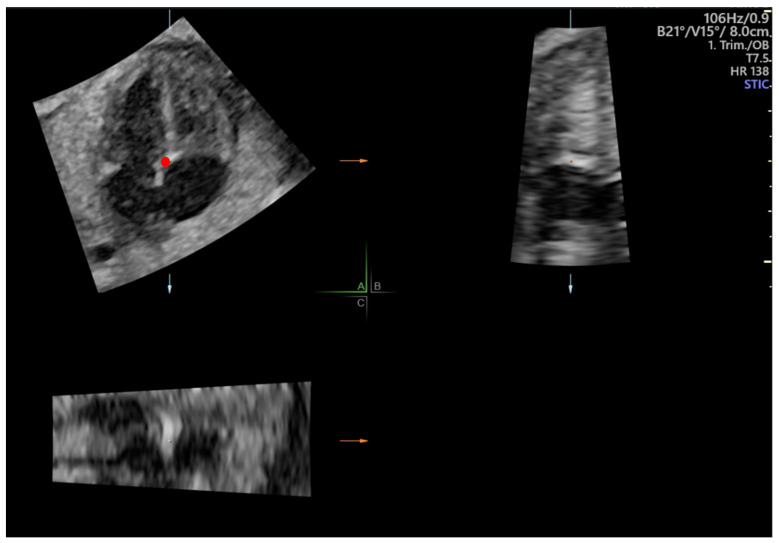
Multiplanar display of a 4D-Spatio Temporal Image Correlation (STIC) volume. In plane A, the reference dot (red dot) is placed at the crux cordis and Z axis rotated until apex is placed at 0°.

**Figure 2 jcm-11-01414-f002:**
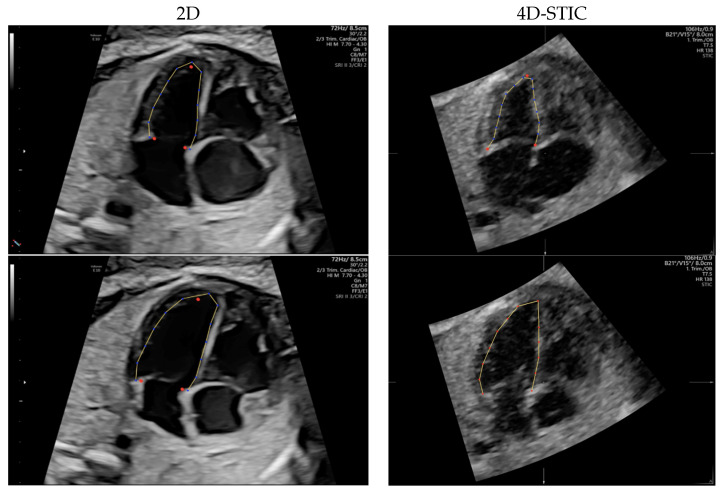
Speckle tracking analysis in a 2-Dimensional (2D) (**left**) and 4D-Spatio Temporal Image Correlation (4D-STIC) volume (**right**) using Fetal Heart Quantification (FetalHQ^®^). Four chamber view in the end systole with the tree reference dots at the septal and lateral mitral valve annulus and at the apex, and tracking of the endocardial border (superior). Tracking of the left ventricle endocardial border at the end diastole (inferior).

**Figure 3 jcm-11-01414-f003:**
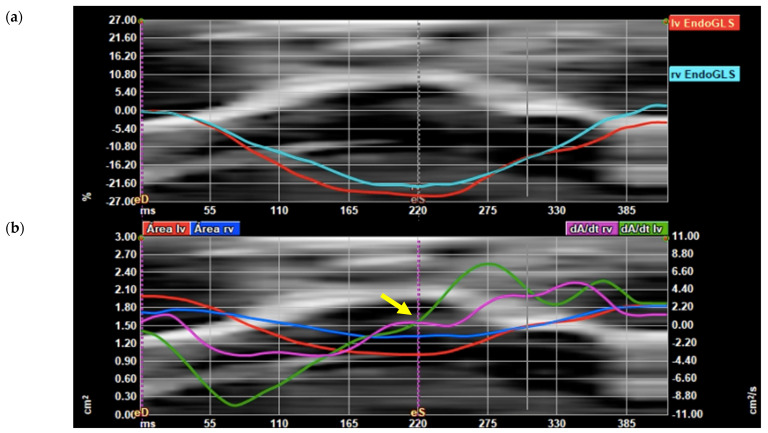
FetalHQ^®^ graphic display. (**a**) Graphic display of left ventricle (LV) global longitudinal strain (GLS) (red line) and right ventricle (RV) GLS (light blue line) with superimposed anatomic M-mode representing tricuspid annulus; (**b**) graphic display of the derivative of the area (dA/dt) of the LV (green line) and the RV (pink line), LV and RV area (red and blue lines, respectively). Systolic frame (eS) is adjusted in the area derivate as a function of the time derivate graph at the time the function crosses 0 (yellow arrow).

**Figure 4 jcm-11-01414-f004:**
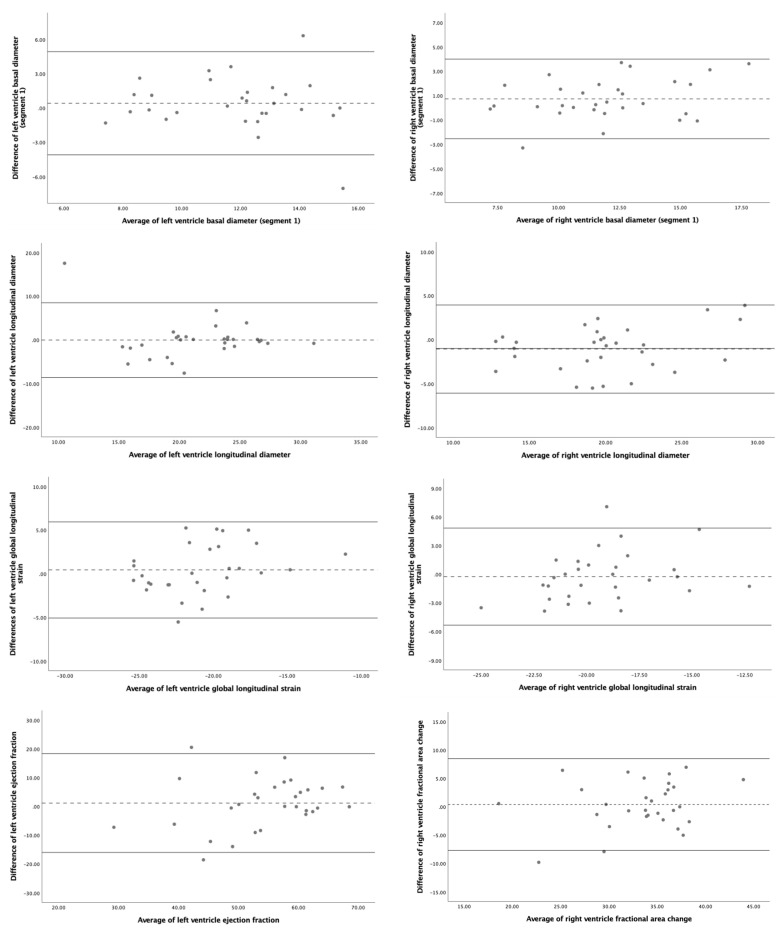
Bland–Altman plots for interobserver reproducibility of fetal heart speckle tracking analysis of morphometric and functional parameters using 4D-Spatio Temporal Image Correlation (4D-STIC). **Upper figures**: left ventricle basal diameter (segment 1) (left) and right ventricle basal diameter (segment 1) (right). **Middle upper figures**: left ventricle longitudinal diameter (left) and right ventricle longitudinal diameter (right). **Middle bottom figures**: left ventricle global longitudinal strain (left) and right ventricle global longitudinal strain (right). **Bottom figures**: left ventricle ejection fraction (left) and right ventricle fractional area change (right).

**Figure 5 jcm-11-01414-f005:**
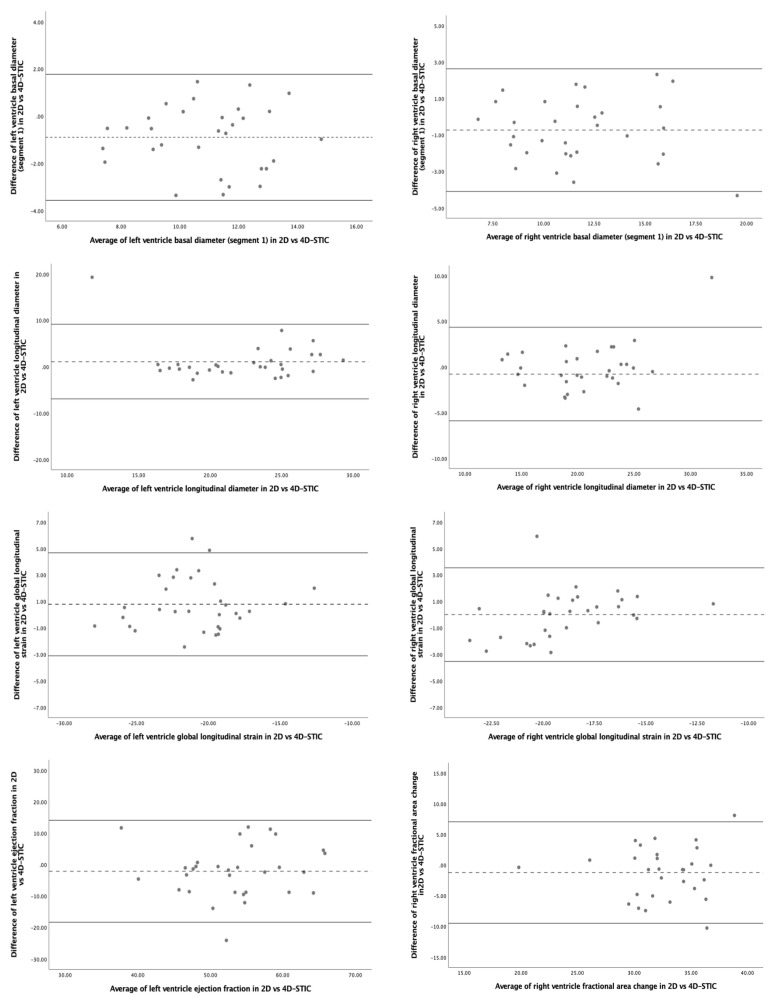
Bland–Altman plots for interobserver reproducibility of fetal heart speckle tracking analysis of morphometric parameters with 2D- versus 4D-Spatio Temporal Image Correlation (4D-STIC). **Upper figures**: left ventricle basal diameter (segment 1) (left) and right ventricle basal diameter (segment 1) (right). **Middle upper figures**: left ventricle longitudinal diameter (left) and right ventricle longitudinal diameter (right). **Middle bottom figures**: left ventricle global longitudinal strain (left) and right ventricle global longitudinal strain (right). **Bottom figures**: left ventricle ejection fraction (left) and right ventricle fractional area change (right).

**Table 1 jcm-11-01414-t001:** Maternal and perinatal characteristics of the study population.

Variable	Result
MATERNAL BASELINE CHARACTERISTICS	
Maternal age, years	33.3 ± 6.16
Body mass index, kg/m^2^	22.1 ± 2.5
Chronic diseases (hypothyroidism, ulcerative colitis)	2 (6.4%)
Race	
White	28 (80%)
Latin American	2 (5.7%)
Asian	1 (2.9%)
Smoking habit	1 (3.2%)
Nulliparity	18 (58.1%)
Use of artificial reproductive technologies	2 (6.5 %)
PERINATAL RESULTS	
Gestational age at birth, weeks	39.5 ± 1.1
Cesarean section	3 (9.7%)
Birthweight, g	3513 ± 417
Birthweight centile	53.7 ± 28
Five minutes APGAR score below 7	0 (0%)
Data expressed as mean ± standard deviation, median (range) or *n* (%)	

**Table 2 jcm-11-01414-t002:** Intraobserver reproducibility of the fetal heart speckle tracking analysis results using 4D-STIC.

Variable	ICC	95% Confidence Interval	*p*-Value	ICC	95% Confidence Interval	*p*-Value
FETAL CARDIAC MORPHOMETRY						
	Left Ventricle			Right Ventricle		
Ventricular Area	0.976	0.950 to 0.988	<0.001	0.970	0.936 to 0.986	<0.001
Longitudinal diameter	0.933	0.862 to 0.968	<0.001	0.970	0.938 to 0.9585	<0.001
Basal diameter (segment 1)	0.853	0.696 to 0.929	<0.001	0.855	0.702 to 0.930	<0.001
Mid-ventricular diameter (segment 9)	0.924	0.841 to 0.964	<0.001	0.936	0.867 to 0.969	<0.001
Apical diameter (segment 17)	0.912	0.818 to 0.958	<0.001	0.943	0.881 to 0.972	<0.001
Basal sphericity index (segment 1)	0.440	−0.201 to 0.736	0.067	0.526	−0.042 to 0.872	0.061
Mid-ventricular sphericity index (segment 9)	0.702	0.392 to 0.855	<0.001	0.665	0.298 to 0.840	0.002
Apical sphericity index (segment 17)	0.787	0.561 to 0.897	<0.001	0.609	0.129 to 0.812	0.006
FETAL CARDIAC FUNCTION						
	Left Ventricle			Right Ventricle		
Global longitudinal strain	0.906	0.807 to 0.955	<0.001	0.732	0.437 to 0.873	<0.001
Fractional area change	0.845	0.665 to 0.926	<0.001	0.746	0.482 to 0.877	<0.001
Basal shortening fraction (segment 1)	0.302	−0.561 to 0.684	0.188	0.775	0.526 to 0.895	<0.001
Mid-ventricular shortening fraction (Segment 9)	0.748	0.472 to 0.879	<0.001	0.801	0.579 to 0.906	<0.001
Apical shortening fraction (segment 17)	0.805	0.599 to 0.906	<0.001	0.619	0.188 to 0.820	0.007
End-diastolic volume	0.968	0.933 to 0.985	<0.001			
End-systolic volume	0.936	0.866 to 0.969	<0.001			
Ejection fraction	0.760	0.501 to 0.885	<0.001			
Cardiac Output	0.782	0.500 to 0.904	<0.001			

ICC: Intraclass Correlation Coefficient.

**Table 3 jcm-11-01414-t003:** Interobserver reproducibility of the fetal heart speckle tracking analysis results using 4D-STIC.

Variable	ICC	95% Confidence Interval	*p*-Value	ICC	95% Confidence Interval	*p*-Value
FETAL CARDIAC MORPHOMETRY						
	Left Ventricle			Right Ventricle		
Ventricular area	0.931	0.857 to 0.967	<0.001	0.966	0.930 to 0.984	<0.001
Longitudinal diameter	0.756	0.483 to 0.885	<0.001	0.909	0.797 to 0.958	<0.001
Basal diameter (segment 1)	0.746	0.464 to 0.881	<0.001	0.891	0.751 to 0.950	<0.001
Mid-ventricular diameter (segment 9)	0.841	0.666 to 0.924	<0.001	0.921	0.835 to 0.962	<0.001
Apical diameter (segment 17)	0.884	0.675 to 0.925	<0.001	0.882	0.745 to 0.994	<0.001
Basal sphericity index (segment 1)	0.390	−0.161 to 0.694	0.064	0.333	−0.227 to 0.659	0.095
Mid-ventricular sphericity index (segment 9)	0.495	−0.11 to 0.754	0.027	0.683	0.329 to 0.850	0.001
Apical sphericity index (segment 17)	0.445	−0.101 to 0.728	0.047	0.628	0.212 to 0.823	0.005
FETAL CARDIAC FUNCTION						
	Left Ventricle			Right Ventricle		
Global longitudinal strain	0.825	0.634 to 0.916	<0.001	0.767	0.508 to 0.889	<0.001
Fractional area change	0.831	0.646 to 0.920	<0.001	0.843	0.671 to 0.925	<0.001
Basal shortening fraction (segment 1)	0.116	−0.951 to 0.595	0.378	0.506	−0.055 to 0.772	0.036
Mid-ventricular shortening fraction (Segment 9)	0.742	0.466 to 0.875	<0.001	0.666	0.303 to 0.839	0.002
Apical shortening fraction (segment 17)	0.782	0.554 to 0.894	<0.001	0.745	0.476 to 0.876	<0.001
End-diastolic volume	0.872	0.718 to 0.942	<0.001			
End-systolic volume	0.773	0.497 to 0.898	<0.001			
Ejection fraction	0.769	0.516 to 0.890	<0.001			
Cardiac Output	0.602	0.082 to 0.828	0.016			

ICC: Intraclass Correlation Coefficient.

**Table 4 jcm-11-01414-t004:** Comparison of Interobserver reproducibility of the fetal heart speckle tracking analysis results using 2D- vs. 4D-STIC.

Variable	ICC	95% Confidence Interval	*p*-Value	ICC	95% Confidence Interval	*p*-Value
CARDIAC MORPHOMETRY						
	Left Ventricle			Right Ventricle		
Ventricular area	0.930	0.817 to 0.970	<0.001	0.949	0.895 to 0.975	<0.001
Longitudinal diameter	0.745	0.480 to 0.876	<0.001	0.907	0.806 to 0.955	<0.001
Basal diameter (segment 1)	0.833	0.529 to 0.930	<0.001	0.912	0.803 to 0.959	<0.001
Mid-ventricular diameter (segment 9)	0.858	0.461 to 0.947	<0.001	0.825	0.638 to 0.915	<0.001
Apical diameter (segment 17)	0.871	0.661 to 0.944	<0.001	0.871	0.753 to 0.938	<0.001
Basal sphericity index (segment 1)	0.222	−0.401 to 0.601	0.211	0.637	0.226 to 0.823	0.003
Mid-ventricular sphericity index (segment 9)	0.463	−0.062 to 0.737	0.018	0.738	0.457 to 0.873	<0.001
Apical sphericity index (segment 17)	0.589	0.172 to 0.799	0.005	0.698	0.370 to 0.855	0.001
CARDIAC FUNCTION						
	Left Ventricle			Right Ventricle		
Global longitudinal strain	0.898	0.779 to 0.952	<0.001	0.878	0.746 to 0.941	<0.001
Fractional area change	0.682	0.350 to 0.846	0.001	0.667	0.307 to 0.842	0.002
Basal shortening fraction (segment 1)	0.643	0.227 to 0.838	0.003	0.339	−0.521 to 0.710	0.163
Mid-ventricular shortening fraction (Segment 9)	0.387	−0.196 to 0,695	0.078	−0.05	−1.1 to 0.493	0.553
Apical shortening fraction (segment 17)	0.501	−0.021 to 0.758	0.030	0.480	−0.091 to 0.752	0.043
End-diastolic volume	0.936	0.669 to 0.978	<0.001			
End-systolic volume	0.896	0.774 to 0.952	<0.001			
Ejection fraction	0.628	0.243 to 0.819	0.004			
Cardiac Output	0.562	−0.149 to 0.826	0.001			

ICC: Intraclass Correlation Coefficient.

## Data Availability

Study data can be made available upon documented request.

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
