# Peer review of "Feasibility of 4D-Spatio Temporal Image Correlation (STIC) in the Comprehensive Assessment of the Fetal Heart Using FetalHQ®"

_jcm, 2022, doi:10.3390/jcm11051414_

Round 1
Reviewer 1 Report
Minor comments
The paper is well written with generally good use of English. The sentence “The current study has several strengths and limitations” is unnecessary and should be removed.
Major points
As the study has been done across a range of gestations it’s a shame the authors didn’t use Bland Altman plots as well as ICC as these would have demonstrated differences in agreement across gestation. I would be interested to see of these could be added for some of the more significant parameters. This would then involve relegating some of the ICCs to supplemental tables which I think would make the papers results easier to interpret. Bland Altman’s would also allow assessment of training effect as the operators did more of these assessments unless they have shown an initial training dataset previously.
The conclusion references the other reports well, but doesn’t adequately explain why some of the parameters have much poor agreement that others and I was left wondering whether it is a fault of the automated software or a result of the type of measurement and also whether this was important. Could the authors attempt to improve on this.
There is also a reference on 312 to EFW only being known in 24 cases which I didn’t notice being mentioned earlier. Does this mean that there are missing data points within the dataset and that all calculations have not been done on 31 participants. If so this requires more explanation and detailing of which analyses have which n numbers
Author Response
Minor comments
The paper is well written with generally good use of English. The sentence “The current study has several strengths and limitations” is unnecessary and should be removed.
RESPONSE:
We appreciate your comment and we have removed the sentence from the text.
Major points
As the study has been done across a range of gestations it’s a shame the authors didn’t use Bland Altman plots as well as ICC as these would have demonstrated differences in agreement across gestation. I would be interested to see of these could be added for some of the more significant parameters.
Thank you very much for this important point. As suggested, we have included the Bland Altman plots for interobserver reproducibility of 4D-STIC and 2D vs 4D-STIC for the most relevant parameters (Figures 4, 5). In order not to exceed the number of images in the main document, we have added the intraobserver reproducibility plots in the supplementary material (Figure S1). However, if the reviewer or the editor think it is better to add all the Bland Altman figures in the main document, we have no inconvenience to modify it. As seen on the plots, gestational age had no impact on our reproducibility analysis. We also assessed agreement with linear regression and t test and no significant differences were observed.
To better clarify this aspect we have made some changes in the text:
“Agreement between operators was studied with Student t test. Limits of agreement, standard error and 95% limit of agreement were calculated and Bland-Altman plots were obtained” (see lines 190-192 in Statistical Analysis).
“No statistically significant differences between operators were observed neither systematic bias for the studied parameters” (see lines 230-232 and 269-271 in Results).
This would then involve relegating some of the ICCs to supplemental tables which I think would make the papers results easier to interpret.
Thank you for the suggestion, but we would like to keep the ICC information on the main text, since part of the discussion focuses on the reproducibility comparing regional vs. global cardiac parameters. We think it will be easier for the readers if they can find all the parameters together in the tables of the main document.
Bland Altman’s would also allow assessment of training effect as the operators did more of these assessments unless they have shown an initial training dataset previously.
Yes, you are right. We did not mention anything regarding training prior to applying a new technology and it is very relevant. We have added the following sentence: “All operators performed a learning curve of more than 20 fetuses prior to the study” lines 162-163 in Methods.
Regarding training effect, we added the following sentence “All operators performed a learning curve of more than 20 fetuses prior to the study” in lines 162-163.
The conclusion references the other reports well, but doesn’t adequately explain why some of the parameters have much poor agreement that others and I was left wondering whether it is a fault of the automated software or a result of the type of measurement and also whether this was important. Could the authors attempt to improve on this.
Thank you for the suggestion. We have made some changes that better explain differences in reproducibility.
“We hypothesize that manual adjustment applied for improving the delineation of both ventricles, especially the RV, led to differences in the endocardial delineation, in particular variations of transverse diameters that compute for the SI and FS segmental analysis. In our study we performed a manual adjustment of the semi-automatic tracking of both ventricles in most of our cases, especially for the RV, considering the endocardium and moderator band as part of the ventricular cavity. Furthermore, SI and FS are calculated by a mathematical formula (SI: end-diastolic longitudinal diameter/end-diastolic transverse diameter, for each segment; FS: (end-diastolic transverse diameter – end-systolic transverse diameter)/end-diastolic transverse diameter, for each of the 24 segments) which may increase the error of both measures and explain the poorer reproducibility compared to other parameters that do not apply formulas. Finally, different orientation of the 4-chamber view acquisition (apical vs transverse) between studies could also explain discrepant results. It is important to note that due to lateral resolution echographic properties, some of the echos of the lateral walls and the upper part of the septum can be missed when analyzing an apical compared to a transverse 4-chamber view. Further studies are necessary to better define these methodological issues and its impact on STE reproducibility of biventricular segmental morphometric evaluation” (Lines: 357-374).
There is also a reference on 312 to EFW only being known in 24 cases which I didn’t notice being mentioned earlier. Does this mean that there are missing data points within the dataset and that all calculations have not been done on 31 participants. If so this requires more explanation and detailing of which analyses have which n numbers
Yes, you are right. The cardiac output parameter is the only one in which we only have 24 fetuses to evaluate the reproducibility. The reason for this is that we didn’t calculate EFW at the moment of the echocardiography if it was already assessed <14 days prior our study, following local guidelines. This happened in 7/31 fetuses. As you know, to calculate cardiac output (ml/min/kg) using FetalHQ®, the EFW must be calculated in the same ultrasound examination, we can’t add later a previous EFW.
We have clarified this with the following sentences: lines 88-89: “Estimated fetal weight was calculated, according to Hadlock et al, in cases in which it was not available in the two weeks prior to the fetal echocardiography” and lines 152-154: “CO was only assessed in 24 fetuses rather than 31, as for the rest of the parameters, since the measurement of estimated fetal weight in the same scan is necessary to obtain the result”.
Additionally we have added this important point as a limitation of our study (See line 423). “Firstly, we are aware that the sample size of our study could be a weakness, especially for the assessment of CO as we only included 24 fetuses”.

Reviewer 2 Report
The paper is very well written, the subject is interesting and novel. It opens up a new area of research for the use of a recently introduced software FetalHQ for analysis of 2D but also 4D STIC images of fetal hearts in terms of morphometry and function.
4D-STIC is found feasible, reproducible and comparable to 2D echocardiography for the assessment of morphometry and function of the heart when FetalHQ is used and when the process is performed by trained and experienced professionals.
And, although this study was on healthy fetuses in the second, beginning of third trimester of pregnancy in the BCNatal, I look forward to the results in fetuses with anomalies (structural or other pathologies like FGR) and on results on telemedicine application.
Author Response
Thank you very much for your comments, we really appreciate them. We are now conducting a study in fetuses with Aortic and Pulmonary stenosis, on FetalHQ® application for the analysis of morphometric and cardiac function. We hope to publish the results very soon.
